# Surface properties of the seas of Titan as revealed by Cassini mission bistatic radar experiments

Valerio Poggiali [1] ✉, Giancorrado Brighi [2], Alexander G. Hayes[1], Phil D. Nicholson[1], Shannon MacKenzie [3], Daniel E. Lalich[1], Léa E. Bonnefoy[1,4], Kamal Oudrhiri[5], Ralph D. Lorenz [3], Jason M. Soderblom [6], Paolo Tortora [2] & Marco Zannoni [2]

Saturn's moon Titan was explored by the Cassini spacecraft from 2004 to 2017. While Cassini revealed a lot about this Earth-like world, its radar observations could only provide limited information about Titan's liquid hydrocarbons seas Kraken, Ligeia and Punga Mare. Here, we show the results of the analysis of the Cassini mission bistatic radar experiments data of Titan's polar seas. The dual-polarized nature of bistatic radar observations allow independent estimates of effective relative dielectric constant and small-scale roughness of sea surface, which were not possible via monostatic radar data. We find statistically significant variations in effective dielectric constant (i.e., liquid composition), consistent with a latitudinal dependence in the methane-ethane mixing-ratio. The results on estuaries suggest lower values than the open seas, compatible with methane-rich rivers entering seas with higher ethane content. We estimate small-scale roughness of a few millimeters from the almost purely coherent scattering from the sea surface, hinting at the presence of capillary waves. This roughness is concentrated near estuaries and inter-basin straits, perhaps indicating active tidal currents.

The Cassini spacecraft explored Saturn's largest moon, Titan, from 2004 to 2017, revealing an Earth-like world with a diverse set of strange, yet very familiar, surface morphologies[1] shaped by a methane-based hydrologic system e.g., refs. [2–4], operating in a dense nitrogen atmosphere e.g., refs. [5,6]. Winds in the lower atmosphere move sediments and shape them into vast dune fields that encircle Titan's equatorial latitudes e.g., refs. [7,8]. In the mid-latitudes, flat and relatively featureless plains[9] mark a transition between the eolian-dominated equator and lacustrine-dominated poles[10]. In the polar regions, large seas and small lakes of liquid hydrocarbons dominate the terrain[11]. Precipitation-fed channels flow into the seas creating estuaries, in some cases deltas, and other familiar coastal sedimentary deposits[12]. While Cassini has revealed much about Titan, these discoveries have prompted more questions[13].

Many of the findings we mentioned above have been made possible by the Cassini RADAR instrument[14], which has explored the morphology of Titan's coastlines using its Synthetic Aperture Radar (SAR) imaging mode and was used as a sounder to probe the depth of Titan's seas up to about 200 m using its altimetry mode. Taking advantage of the Ku-band ($\lambda_{Ku} = 2.17$ cm) transparency of liquid hydrocarbons, the radar altimeter was able to provide some insight into the seas' composition by examining microwave absorption as a function of sea depth. The most probable loss tangents of observed liquid columns are about $10^{-5}$[15]. Such a low value is consistent with ternary mixtures of methane, ethane, and nitrogen, though with less ethane than predicted[3]. Individual measurements, however, are affected by significant errors (1σ of about 20–40%). Moreover, the

[1]Cornell University, Ithaca, NY, USA. [2]Università di Bologna, Dipartimento di Ingegneria Industriale, Forlì, Italy. [3]JHU / APL, Laurel, MD, USA. [4]LERMA, Observatoire de Paris, CNRS, Paris, France. [5]NASA Jet Propulsion Laboratory, California Institute of Technology, Pasadena, CA, USA. [6]MIT, Cambridge, MA, USA. ✉e-mail: vpoggiali@astro.cornell.edu

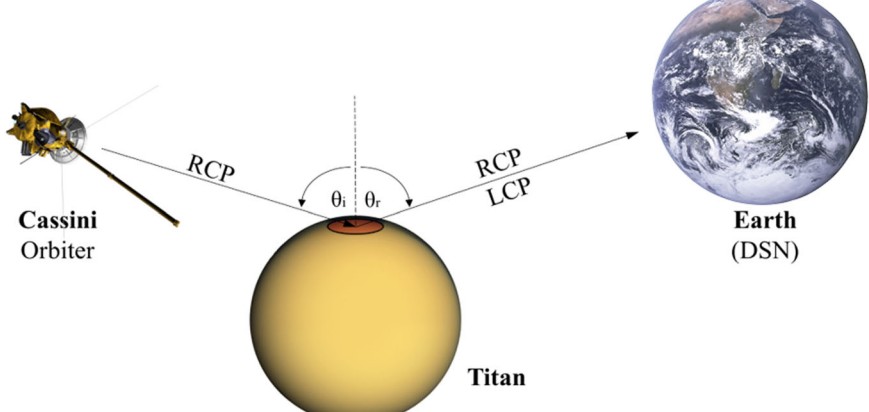

**Fig. 1 | Observing geometry and signal path for Cassini bistatic radar downlink experiments.** Cassini's antenna is pointed at the predicted specular point where the angle of incidence ($\theta_i$) equals the angle of reflection ($\theta_r$), measured from the local vertical. Right circularly polarized (RCP) signals were transmitted, and at least one of the NASA Deep Space Network (DSN) antennas on Earth captured the echo. At the surface of Titan, a left circularly polarized (LCP) component of the echo signal was generated, requiring the presence on the ground of two receiving channels. Complex samples of each receiver output were collected at 16,000 samples/s and stored for later processing. The purpose of this figure is to illustrate the geometry of the observations and distances on the coordinate axes that are not drawn to scale.

interpretation of surface and seafloor backscatter was further complicated by the difficulty of separating the effect of microscale roughness from the relative dielectric constant. Regardless, the altimetry results suggested that Punga Mare should have a lower ethane content (i.e., lower loss tangent) than Ligeia Mare and Moray Sinus[16,17]. Unfortunately, no direct estimate of the composition and/or depth of central Kraken was possible as seabed echoes were not detected[16,18]. Note that a latitudinal gradient in ethane content was predicted by ref. 19 in response to the expected latitudinal gradient in precipitation frequency and expected evaporation rates. As rain on Titan is almost entirely methane/nitrogen, the northernmost lakes and seas may be flushed with methane and have a lower ethane content, much as 'fresh' continental runoff flushes saltwater from the Baltic into the Atlantic, or from the Sea of Azov into the Black Sea and thence the Mediterranean.

In an attempt to refine and expand our understanding of how Titan's seas and atmosphere interact, here we bring to bear a previously unexploited dataset. Between March 2006 and November 2016, Cassini conducted 13 bistatic radar (BSR) observations of Titan's surface utilizing its Radio Science Subsystem (RSS). The spacecraft transmitted monochromatic right circularly polarized (RCP) signals at S ($\lambda_S = 13$ cm), X ($\lambda_X = 3.56$ cm), and Ka ($\lambda_{Ka} = 0.9$ cm) band. As shown in Fig. 1, when an RCP wave interacts with a gently undulating planetary surface, reflected echoes are expected to travel mostly in the specular direction, and show two orthogonally polarized components (RCP and LCP), whose power ratio is determined by the near-surface dielectric constant and the geometry of the observation[20]. The full scattering matrix is reconstructed on Earth from signals recorded by two (coherent) open-loop receivers at one of the deep space network (DSN) stations (i.e., Canberra, Goldstone, or Madrid). Surface roughness on length scales of hundreds of wavelengths (meters) is expected to broaden the reflected signal in frequency due to Doppler shifts induced by tilted surface facets distributed over the reflecting area[20].

The analysis of the RSS bistatic data acquired over Titan's seas offers a unique opportunity to constrain and validate the surface and liquid column composition information inferred from Cassini's Ku-Band RADAR altimeter[1,15–17,21–23] and radiometer[24] observations. Combining bistatic measurements with the context provided by other Cassini data is key for refining previous measurements and producing new and important constraints on the composition and texture of Titan's surfaces.

Herein, we present a brief overview of the Cassini RSS bistatic dataset used for this study and provide independent estimates of

effective relative dielectric constant and small-scale roughness of Titan's sea surfaces, achieved by exploiting the dual-polarized nature of BSR experiments. A statistically significant increase in effective dielectric constant with decreasing latitude is observed. Results also suggest that coastal areas, like estuaries, are characterized by increased roughness and a lower effective dielectric constant than the open sea. The results obtained are then placed in context with other Cassini observations to constrain the composition, surface properties, and dynamics of Titan's polar seas.

## Results

In this paper, we analyze four bistatic observations of Titan carried out by Cassini RSS during flybys T101 (05/17/2014), T102 (06/18/2014), T106 (10/24/2014), and T124 (11/14/2016). Each flyby consisted of an ingress bistatic observation preceding Cassini's closest approach to Titan, followed by an egress bistatic observation. When the closest approach occurred behind Titan (with respect to Earth), a radio occultation experiment was conducted between ingress and egress (i.e., T101 and T102). We limit our analysis to the egress phase of these four flybys because these are the only observations whose specular point crossed the main body of at least one of the three polar seas (Kraken, Ligeia, and Punga Mare). For this work we used only X-band data acquired by the 70 m Canberra DSN station because of its higher SNR compared to the other frequency bands.

The first observation over a sea was collected during flyby **T101**, when the specular point traversed Ligeia Mare (L), briefly transited Moray Sinus (MS), and then ended within the northern part of Kraken Mare (K1A near the coast, K1B in open sea). See Fig. 2. During **T102** the specular track covered Ligeia, crossed Trevize Fretum (TF), before moving on to the Moray Sinus estuary and the main body of northern Kraken Mare (K2), ending in the Bayta Fretum strait (K3). **T106** traversed the southeastern portion of Kraken Mare (K7), offering the first chance to probe the composition of southern Kraken, then crossed central Kraken (K5), finally ending in the eastern portion of northern Kraken (K4). **T124** crossed the south-west area of Punga Mare (P), Dingle Sinus and Genova Sinus (GS1 and GS2), before entering the main body of the center-western Kraken Mare (K6) and then ending in south-western Kraken Mare (K8).

The raw RSS data for the two orthogonally polarized channels were processed and calibrated to extract frequency spectra of reflections from Titan's surface (see the Methods section and sample code

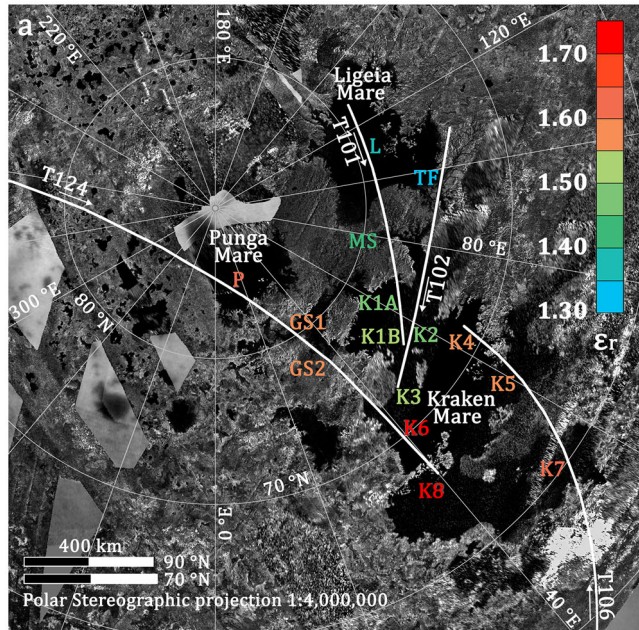

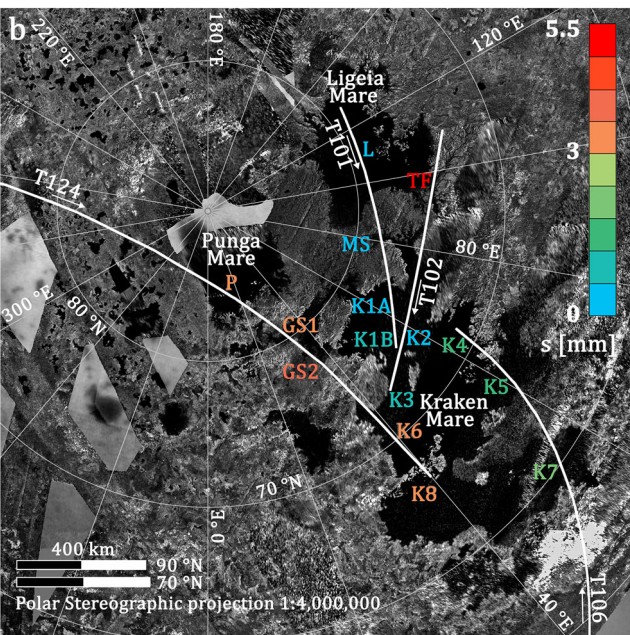

**Fig. 2 | Specular point tracks of the bistatic observations during experiments T101, T102, T106, T124.** Tracks are plotted on a RADAR/ISS basemap mosaic showing the Titan's three large polar liquid hydrocarbon seas, Kraken Mare (60–80 °N, 30–80 °E), Ligeia Mare (75–83 °N, 80–140 °E), and Punga Mare (83–87 °N, 0–60 °E). Labels indicate those areas where the surface's effective **a** average relative dielectric constant ($\varepsilon_r$) and **b** small-scale rms height (s) roughness have been measured and are color-coded with the retrieved values (see Results section and the Source data that are provided as a Source Data file).

(e.g., about 0.01 Hz/s for T101). Although weak diffuse scattering components may be present in the total received echo, the predominant features are the powerful mirror-like reflections coming from the sea surface, with magnitudes that are 10–100 times higher than previously recorded during bistatic experiments on Mars[20] and Venus[29]. Analysis was limited to echoes with a returned signal SNR >5 dB for which the Fresnel footprint consisted solely of liquid surfaces far from coastal areas (larger than about 20 km).

Under certain geometrical conditions, only part of the surface capable of contributing to the echo may be illuminated by the antenna pattern. This could eventually cause the received power to be lower than that expected. The almost monochromatic nature of the echoes received during the X-band bistatic experiments over the seas suggests that nearly all the reflected signal came from an elliptical Fresnel zone with axes of less than $1 \times 2$ km[30]. Given that the angular separation between Cassini's high gain antenna boresight direction and the theoretical specular ray was generally very small (i.e., <0.1° for T101 vs a beam size of 0.635° at X-band[31]), we can confidently say that the antenna was always able to fully illuminate the region encompassing the specular return[31].

Titan's seas are believed to be dominantly mixtures of ethane ($\varepsilon_r \cong 2$), methane ($\varepsilon_r \cong 1.73$), and dissolved nitrogen ($\varepsilon_r \cong 1.5$) [22,27,32–34]. We derived values of the average effective relative dielectric constant ($\varepsilon_r$) from the incidence angle θ and circular polarization ratio (CPR)[35] (see Methods section for more details) for several areas of the seas (see Fig. 2, Table 1, and Supplementary Fig. S1). The map of $\varepsilon_r$ shows a slight, but statistically significant, increase with decreasing latitude. Ligeia Mare shows statistically significant lower values in effective dielectric constant than Kraken Mare, especially when compared with the central and southern portions of Kraken. While the main body of Ligeia Mare (L) exhibits an average value of $1.38 \pm 0.03$, the central portion of Kraken Mare (K6) exhibits a value of $1.71 \pm 0.11$. Differences in uncertainties in the $\varepsilon_r$ reported are primarily due to variations in noise level and/or heterogeneities in the sea surface.

Observations K1B and K2 were acquired in two areas of the northern Kraken Mare very close to each other during T101 and T102, with similar sea-state conditions (roughness) and incidence angles of acquisition, and resulted in dielectric constants of $1.52 \pm 0.1$ ($\theta = 61.3°$) and $1.45 \pm 0.05$ ($\theta = 63.7°$), respectively, indistinguishable within 1σ. The agreement between separate observations of the same area supports the reliability of our calibration procedure. The 3-min-long T124 observation of the surface of Punga Mare resulted in values of $\varepsilon_r = 1.64^{+0.07}_{-0.09}$ similar to those recorded in the southern portion of Kraken Mare (i.e., K7). Previous measurements by the Cassini RADAR altimeter of Punga Mare (T108, acquired over the southeastern area of the sea in Jan. 2015) found a most probable real part of the dielectric constant of the mixture of $1.67 \pm 0.05$[17].

Compared to central and southern Kraken, the river estuaries (Trevize Fretum, Genova Sinus, and Moray Sinus) show lower values of average effective dielectric constant, which may be associated with mixtures of liquid hydrocarbons with lower ethane-methane mixing ratios and/or with a surface characterized by higher porosity/lower density. Estuaries are where we would expect methane-rich rivers to enter more ethane-rich seas[18], creating an environment analogous to brackish water found at the intersection of freshwater rivers and salt-water seas on Earth.

As a general remark, we also report the possible presence of a moderate correlation between the estimates of $\varepsilon_r$ and incidence angle. Pearson's correlation coefficient, which describes the strength of the linear association between the two variables, is equal to $-0.60^{+0.21}_{-0.15}$ (at 1σ error). The expressions for the Fresnel reflection coefficients account for the angle of observation when relating relative dielectric constant and reflectivity, but far from the Brewster angle the power computation in the weaker polarization introduces uncertainty in the $\varepsilon_r$ estimates. Still, no systematic bias should be present[36]. In this regard,

included in the "Supplementary Data 1" archive). Echoes received from the seas are extremely narrow (see Fig. 3 and ref. 25). Large-scale surface roughness produces a broadening of the received echoes, but in this case, after using an integration time of 1.024 s, we find a spectral width of the order of the frequency resolution (about 1 Hz), suggesting extremely smooth surfaces on a length-scale of hundreds of wavelengths (see also refs. 26–28). This property of the received echoes simplifies the analysis by permitting easier tracking and an eventual correction of residual Doppler drifts recorded during observations

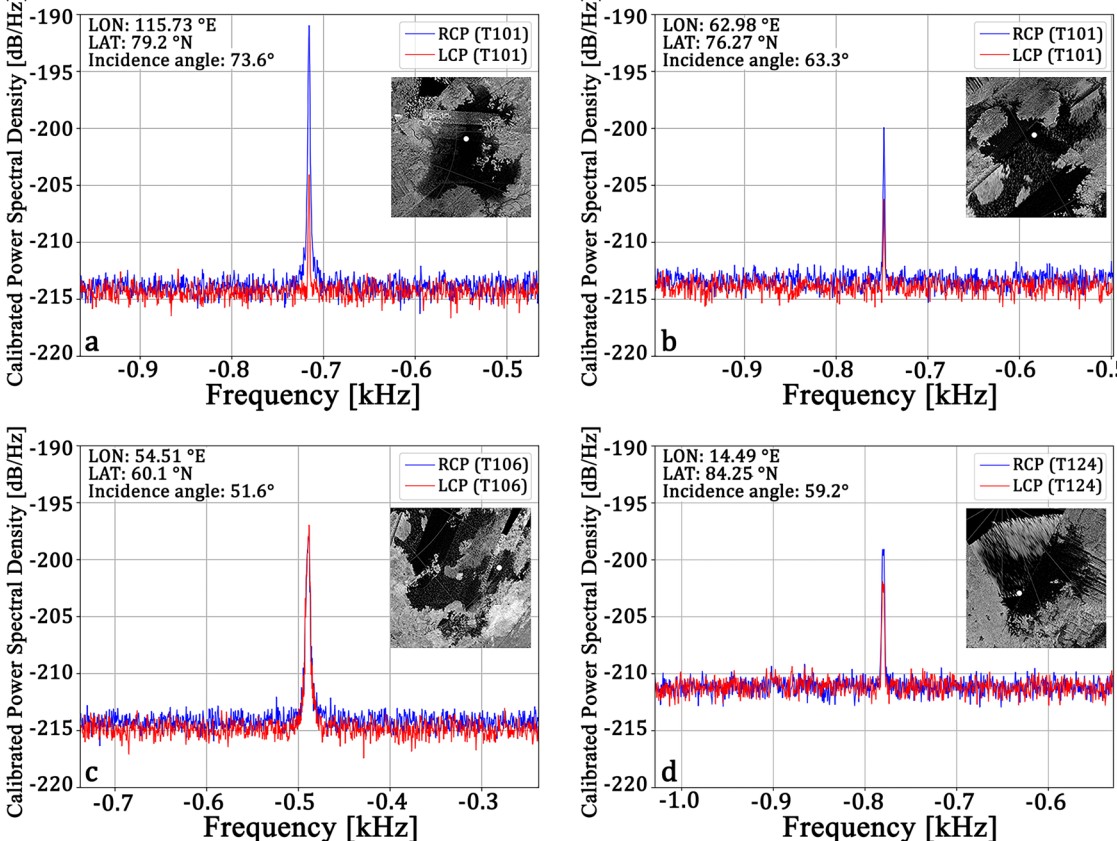

**Fig. 3 | Four calibrated spectra from the three seas of Titan acquired during T101, T106, and T124 observations.** A powerful specular reflection is clearly visible for all of them in both the RCP and LCP polarizations. These echoes have been obtained by integrating roughly 60 s of observation time. **a** Echoes from Ligeia Mare; note how the RCP component dominates over the LCP because the incidence angle here is higher than the Brewster angle (i.e., $\theta_B \cong 52°$ at $\varepsilon_r = 1.6$); inset: radar map of Ligeia Mare with the location of specular reflection (white dot mark) at 115.73 °E, 79.2 °N; **b** Echoes from north Kraken Mare; inset: radar map of north Kraken Mare with the location of specular reflection (white dot mark) at 62.98 °E, 76.27 °N; **c** Echoes from south Kraken Mare; note how the similarity in magnitude between LCP and RCP components is consistent with the vicinity of the incidence angle to the Brewster angle; inset: radar map of north south Kraken Mare with the location of specular reflection (white dot mark) at 54.51 °E, 60.1 °N; **d** Echoes from Punga Mare; note the increase of noise level due to a rain event happening at Canberra during the T124 observation of Punga Mare (see Method section for more details); inset: radar map of Punga Mare with the location of specular reflection (white dot mark) at 14.49 °E, 84.25 °N. All values indicated in the figure and in this caption are given at the midpoint of a 60 s integration interval.

previous observations of Titan's seas made by the Cassini radar altimeter have indicated an effective small-scale rms height (s) to be on the order of a mm[21,34,37]. Experimental investigations conducted on Earth at X-band[38,39] demonstrated that when the electromagnetic surface roughness is very low, as we expect in our case ($2\pi s/\lambda < 0.2$), then the Fresnel reflection coefficient equations adopted in this study for the estimation of $\varepsilon_r$ (see Method section) are in close agreement with measured reflectivities at all incidence angles investigated (from 20° to 65°) and especially above the Brewster angle.

Using the derived average values of surface relative dielectric constant reported in Table 1 as inputs to the perfectly conducting sphere reflection model of Fjeldbo[35] (see Eq. (11) in the Method section), we independently estimated the same values of effective small-scale rms height (s) for both polarization channels and averaged these over each of the areas of interest, reporting also their associated 1σ errors (see Supplementary Fig. S2). Our results are summarized in Fig. 2, Table 1, and Supplementary Fig. S1).

We found that, at the time of the observations, the sea surfaces of the main body of Ligeia, Punga, and Kraken were mostly level, with no major disturbances causing significant disruptions to the surface ($s \leq 3.3$ mm). For some regions observed at high incidence angles during flyby T101 and T102, i.e., L, K1A, and K2, Fjeldbo's model somewhat underestimates the power actually received on the ground (even assuming the sea surface to be perfectly flat). The low values of

relative dielectric constant found for those regions do not justify specular echoes to be as strong as the ones observed (at least under Kirchhoff Approximations). However, note that a slight increase in the relative dielectric constant (<0.06, and thus comparable with estimation errors) would be enough to match the received power if the sea surface was flat at the time of the observation. It could be that Fjeldbo's model for planetary radar cross-section does not accurately describe the scattering processes arising from Titan's sea surface at very high incidence angles. For completeness, we note that an upper bound for s of 1 mm was derived from the analysis of Cassini BSR observations during flybys T101 and T102 by ref. 40, but there is no mention of the reflection model used to carry out the estimate. On the other hand, for lower angles of observation, our predictions match with computed powers, and the inferred values of effective roughness are consistent with past analyses of both monostatic [e.g., 34] and BSR observations[40].

Coastal areas like the Trevize Fretum and the Genova Sinus show increased roughness ($3.6 \leq s \leq 5.2$ mm), a possible indication of surface activity. These areas are believed to represent connections between Ligeia/Kraken and Punga/Kraken, possibly through tidal currents[41–43]. Similar to[16,44], we report the presence of a localized anomalous drop in the RCP received power (about 7 dB) recorded during T102 at the mouth of the Moray Sinus (75.5 °N, 72.5 °E). We cannot say anything about the local surface composition since the signal in the LCP channel

**Table 1 | Estimated average effective relative dielectric constant ($\varepsilon_r$) and small-scale rms height (s) as measured at X-band for a selection of areas of interest in the seas**

| Area | $\varepsilon_r$ | Roughness, s [mm] | Flyby (# of samples) | Sample Integr time [s] | (LAT, LON) [°] | S/C Altitude [km] | Incidence angle, θ [°] |
|---|---|---|---|---|---|---|---|
| L | $1.38^{+0.03}_{-0.03}$ | 0* | T101 (12) | 60 | (79.15, 108.05) | 10556 | 72.3 |
| TF | $1.34^{+0.07}_{-0.08}$ | $5.2^{+1.9}_{-2.7}$ | T102 (11) | 10 | (74.89, 95.97) | 10401 | 72.1 |
| MS | $1.44^{+0.07}_{-0.06}$ | $0^{+0.6}$ | T101 (8) | 10 | (78.32, 80.82) | 14952 | 67.3 |
| GS1 | $1.56^{+0.19}_{-0.18}$ | $3.6^{+0.9}_{-0.7}$ | T124 (6) | 10 | (80.13, 32.51) | 8581 | 58.9 |
| GS2 | $1.57^{+0.21}_{-0.25}$ | $4.0^{+1.1}_{-0.9}$ | T124 (8) | 10 | (78.4, 35.43) | 9687 | 58.8 |
| P | $1.64^{+0.07}_{-0.09}$ | $3.3^{+0.3}_{-0.4}$ | T124 (18) | 10 | (84.74, 8.18) | 6226 | 59.2 |
| K1A | $1.46^{+0.03}_{-0.06}$ | 0* | T101 (17) | 60 | (76.29, 63.17) | 22287 | 63.3 |
| K1B | $1.52^{+0.1}_{-0.1}$ | $1.2^{+0.7}_{-1.2}$ | T101 (24) | 60 | (75.02, 56.07) | 28966 | 61.3 |
| K2 | $1.45^{+0.05}_{-0.05}$ | 0* | T102 (23) | 60 | (74.89, 63.02) | 18920 | 63.7 |
| K3 | $1.53^{+0.13}_{-0.12}$ | $1.0^{+0.5}_{-0.9}$ | T102 (24) | 60 | (73.28, 48.69) | 29252 | 59.7 |
| K4 | $1.58^{+0.07}_{-0.08}$ | $1.8^{+0.6}_{-0.6}$ | T106 (36) | 60 | (70.21, 63.49) | 20578 | 59.1 |
| K5 | $1.59^{+0.02}_{-0.01}$ | $1.6^{+0.2}_{-0.3}$ | T106 (10) | 60 | (66.69, 61.10) | 8453 | 56.9 |
| K6 | $1.71^{+0.11}_{-0.11}$ | $3.2^{+0.4}_{-0.4}$ | T124 (22) | 60 | (68.66, 40.08) | 24298 | 57.8 |
| K7 | $1.62^{+0.04}_{-0.04}$ | $2.0^{+0.2}_{-0.3}$ | T106 (22) | 10 | (60.35, 54.77) | 3306 | 51.8 |
| K8 | $1.71^{+0.21}_{-0.20}$ | $3.1^{+0.5}_{-0.5}$ | T124 (6) | 60 | (66.74, 39.84) | 32084 | 57.5 |

The velocity of the specular point across the surface and the strength of the echo varies with the ever-changing geometry of the observation, and consequently, we had to adjust signal integration time on a case-by-case basis: from 10 s for liquid bodies of limited extent to up to one minute for open seas. Reducing the integration time also avoids possible solid surface contaminations (i.e., during observations crossing river estuaries).

\* In these cases, we have not been able to establish a reliable upper bound on the roughness estimate because Fjeldbo's model underestimates the actual received power. This could be due to a slight underestimate of the relative dielectric constant retrieved over those regions (all were observed with very high incidence angles, i.e., ≥60–70°). Regardless, note that the preliminary analysis made by ref. 40 at the time of the observations was able to establish an upper bound of 1 mm on the measurements of roughness.

is lost. Assuming the same surface composition as the surroundings and accepting small-scale roughness over this limited area (<30 km) as a possible explanation for the drop in power, we found s = 9.3 (±0.2) mm (note that the surrounding shows no appreciable roughness otherwise). Finally, we note that the central and southern portions of Kraken are home to a higher level of roughness with respect to the northern part of the same sea. The southern portion of Kraken is thus characterized by higher values of both effective dielectric constant and small-scale roughness if compared to the more northern seas.

## Discussion

Unlike sounding observations, which interrogate the entire liquid column, bistatic measurements specifically probe the sea's surface layer. Our derived values of the sea surfaces' effective dielectric constants are universally lower (≤ of about 0.2) than expected based on the most probable compositions inferred from liquid column loss tangents[16,17] estimated from Cassini radar altimetry. However, our results are also consistent with the (albeit large) range of dielectric constants inferred from passive radiometry by Cassini at Ligeia Mare. Combining the observed depth profile from ref. 22 with a simple two-layer model[24], found that Ligeia's column-integrated dielectric constant was 1.57 ± 0.25, assuming a rms sea surface height of 1 mm. According to the bistatic observations, the value of $\varepsilon_r$ from Ligeia Mare's surface was $1.38^{+0.03}_{-0.03}$, within the 1-σ uncertainty of the radiometer's estimation. We also found that the small-scale roughness derived from a Physical Optics scattering model[45] using effective $\varepsilon_r$ values from the bistatic experiments for Ligeia Mare (0–1 mm) agrees with roughness values previously derived from RADAR altimeter data (0.1–1.3 mm) using liquid column-integrated $\varepsilon_r$ estimations. For the north portion of Kraken (K1A), the altimeter indicates a rougher surface (2.6–2.9 mm) than retrieved by the bistatic experiment (0–1 mm), although these measurements were taken 3 months apart. For Punga Mare, the altimeter indicated values of 1.4–1.9 mm while the bistatic experiment measured 2.9–3.6 mm about 22 months later. Regardless, the low sea surface dielectric constants implied by the bistatic dataset

are surprising and lower than what would be expected for liquid methane-ethane-nitrogen mixtures ($\varepsilon_r = 1.5$–2). Possible explanations for this discrepancy include calibration errors, a sea surface covered by unconsolidated material, or invalid assumptions in deriving dielectric constants from the observed cross-polarization ratio.

We can generally rule out issues with the absolute calibration algorithms for the reasons listed in the Methods section. If the seas' surfaces were covered by an unconsolidated material (e.g., precipitated from the atmosphere[6]), it could reduce the observed near-surface dielectric constant [16,46–49], similar to what was observed for the uppermost layer of the lunar regolith probed by means of bistatic radar[50] or to what was recorded by Viking at the high northern latitudes of Mars during its high incidence angles bistatic radar observations[51]. In particular, values of $\varepsilon_r$ lower than expected were measured by Viking during the first part of the experiment performed on 25 February 1978, when the bistatic track crossed the region of Planum Boreum. During the transition from the northern polar cap area to the more southern bare soils, an increase in the expected values of $\varepsilon_r$ was found. Radar simulations using the facet method were performed[51] that found little difference between simulated data and predictions based on Fjeldbo's analytical expressions for θ ≤ 82° and rms slopes ≤4°. Such results were explained with the presence of a low-density $CO_2$ or $H_2O$ snow layer increasing its thickness with latitude or, alternatively, with a layer of increasingly porous nonseason material[51].

In the case of Titan, however, it remains unclear how such a layer could be maintained in the presence of precipitation and currents. Furthermore, this would not be consistent with the visual and infrared mapping spectrometer (VIMS) observations of Ontario Lacus that indicate a smooth featureless surface at the micron-scale[27]. Alternatively, Hagfors' assumption[52] that low values of surface roughness have limited influence on the radar cross-section may not hold true in our almost pure coherent scattering regime. If Hagfors' assumption is not valid, our dielectric constant estimation could be affected by the presence of a superficial sub-wavelength-scale roughness. Interestingly, such a roughness would produce the same effect as if a

wavelength-scale density gradient was present at the surface[53]. Accordingly, if Hagfors' assumption is not valid, the presence of wind-generated capillary-gravity waves[44] and/or transient nitrogen exsolutions in the form of bubbles[54–56] at the sea surface could lead to an underestimation of the dielectric constant.

We have analyzed the four Cassini RSS bistatic radar observations of Titan's hydrocarbon seas. The resulting surface effective relative dielectric constant $\varepsilon_r$ and roughness s, show that even under the same sea-state conditions (i.e., roughness), Kraken Mare has a higher effective dielectric constant ($\varepsilon_r$) than Ligeia Mare (1.52 ± 0.1 in K1B vs 1.38 ± 0.03, respectively). We cannot, however, unambiguously distinguish between changes in liquid composition vs. changes in the nature of a surficial layer[57] to explain the observed differences in dielectric constant. Regardless, the southernmost portion of Kraken Mare shows the highest dielectric constant values (see Fig. 2). If the difference is compositional, then these areas could be the most ethane-dominated, consistent with the latitudinal gradient predicted by ref. 19. The values of $\varepsilon_r$ found on the main body of Kraken Mare exhibit a maximum range of variation from 1.45 ± 0.05 (K2) to 1.71 ± 0.11 (K6) going from North to South, with higher values implying greater ethane content. Finally, we found that at the time of the bistatic observations, the main bodies of Ligeia, Punga, and Kraken were mostly level, with no major disturbances causing significant disruptions to the surface (s ≤ 3.3 mm). A higher level of roughness was detected in coastal areas, near estuaries and inter-basin straits, perhaps indicating active tidal currents (3.6 ≤ s ≤ 5.2 mm).

## Methods

The raw RSS data of the two orthogonally polarized channels are sequences of complex samples made of in-phase and quadrature components (IQ samples) of the incoming signal sampled with a fixed frequency (16 kHz for this dataset)[58]. Data processing was performed using a basic averaged periodogram technique, looking for the best trade-off between spatial resolution on the surface, frequency resolution, and SNR. We carried out absolute amplitude calibration of the samples from system noise temperature models alongside an equalization to remove ripples introduced in the noise floor by the open-loop radio science receivers. A Gaussian function was fitted to the calibrated waveforms to measure the spectral broadening of the echoes at their full-width-half-maximum. The total reflected power was obtained in both polarizations by integrating the power spectral density over a frequency band spanning four times the measured signal spectral width[45]. The relative dielectric constant $\varepsilon_r$ of the surface can be derived from the circular polarization ratio (CPR) of the same sense and opposite sense received powers through the Fresnel voltage reflection coefficients. The power ratio can be reduced to a ratio between the surface reflectivity in the two polarizations[36], which depends on the relative dielectric constant of the reflecting surface and the incidence angle of observation ($\theta$).

$$CPR = \frac{P_R}{P_L} = \frac{|R_R|^2}{|R_L|^2} \tag{1}$$

where

$$R_R = \frac{R_V + R_H}{2} \tag{2}$$

and

$$R_L = \frac{R_V - R_H}{2} \tag{3}$$

$R_V$ and $R_H$ are the Fresnel voltage reflection coefficients for horizontal and vertical polarizations, and they can be expressed as

$$R_H = \frac{cos\theta - \sqrt{\varepsilon_r - sin^2\theta}}{cos\theta + \sqrt{\varepsilon_r - sin^2\theta}} \tag{4}$$

and

$$R_V = \frac{\varepsilon\, cos\theta - \sqrt{\varepsilon_r - sin^2\theta}}{\varepsilon\, cos\theta + \sqrt{\varepsilon_r - sin^2\theta}} \tag{5}$$

Inverting this whole system of equations results in

$$\varepsilon_r = \left(\frac{tan^2\theta}{CPR} + 1\right) sin^2\theta \tag{6}$$

For a surface of a given relative dielectric constant, there is a unique incidence angle that yields the circular polarization ratio to unity. This is called the Brewster angle and can be easily retrieved as:

$$\theta_B = arctan\left(\sqrt{\varepsilon_r}\right) \tag{7}$$

Cassini's bistatic observations were usually performed at specular incidence angles $\theta$ ranging 50°–70°, thus slightly larger than the Brewster angle expected for Titan's liquid surfaces ($\varepsilon_r \simeq$ 1.5–2 corresponds roughly to $\theta_B \simeq$ 50°–55°).

Theoretical studies[35] found a relation between the broadening of the received echoes and the large-scale (up to 100 s of $\lambda$[20], which is on the order of meters at X-band) surface roughness in terms of rms-slope for a surface with Gaussian statistics and s ≫ $\lambda$, where s is the root mean square (rms) variation in the surface height. Previous analyses of the Cassini RADAR altimeter and VIMS datasets acquired over Titan's seas didn't reveal any significant surface roughness at large-scale (10 s of km) but only at cm-scales or shorter. These observations were consistent with an upper limit on s of a few mm[16,21,34,37,59], suggesting incredibly smooth surfaces with perhaps the presence of capillary waves at the wavelength-scale ($\lambda_{Ku}$). As a result, the mm-scale roughness of Titan's seas fails to satisfy the assumptions of previous models when compared to an X-band wavelength of 3.6 cm[35]. However, no assumption prevents the CPR of the received echoes from being used to estimate the relative dielectric constant of the surface. Once the effective surface dielectric constant is known, the bistatic scattering coefficient measured from either channel can be used to constrain the surface s at the wavelength scale for mainly coherent surface reflections[39]. While, in general, surface scattering consists of coherent and incoherent components, the flatness of Titan's seas will cause a substantial suppression of the incoherent component. According to the Rayleigh Criterion s<$\lambda$/(8sin$\gamma$), with $\gamma$ the depression angle, reflections from a surface observed at 50°−70° incidence angle can be considered predominantly coherent with s < 1.3 cm at X-band[39]. Unlike single-polarization monostatic RADAR measurements, where one cannot separate the effects of surface microscale roughness from composition[26,37,60], with dual-polarization bistatic RSS data we are able to infer the effective values of both parameters. To investigate the wavelength scale of sea surface roughness, we use the power measured in each of two orthogonally polarized channels. More reliability is accorded to the results obtained using the RCP channel for flyby T101 and T102, due to their higher SNR values acquired at $\theta > \theta_B$. The results that we obtain through the use of the LCP channel give similar (where not identical) outcomes, but with larger errors.

**Table 2 | DSS 43 (Canberra) noise diode temperatures**

| Flyby | $T_{ND}$ (Pre-CAL) X-RCP, K | $T_{ND}$ (Pre-CAL) X-LCP, K | $T_{ND}$ (Post-CAL) X-RCP, K | $T_{ND}$ (Post-CAL) X-LCP, K |
|---|---|---|---|---|
| T101 (05/2014) | 22.88 ± 1.56 | 13.32 ± 1.05 | 23.04 ± 1.48 | 13.22 ± 0.94 |
| T102 (06/2014) | 23.43 ± 1.63 | 13.38 ± 1.00 | 23.26 ± 1.61 | 13.00 ± 1.03 |
| T106 (10/2014) | 23.48 ± 1.56 | 13.08 ± 1.05 | 24.85 ± 1.53 | 13.96 ± 1.23 |
| T124 (11/2016) | 20.99 ± 1.42 | 14.22 ± 1.04 | 21.18 ± 1.46 | 14.28 ± 1.07 |

Values are provided with their associated 1σ errors estimated for the bistatic experiments of interest for this paper during the respective pre- and post-calibration routines.

**Table 3 | DSS 43 (Canberra) inherent receiver temperatures**

| Flyby | $T_{REC}$ (Pre-CAL) X-RCP, K | $T_{REC}$ (Pre-CAL) X-LCP, K | $T_{REC}$ (Post-CAL) X-RCP, K | $T_{REC}$ (Post-CAL) X-LCP, K |
|---|---|---|---|---|
| T101 (05/2014) | 24.67 ± 0.78 | 18.40 ± 0.59 | 24.70 ± 0.81 | 18.44 ± 0.55 |
| T102 (06/2014) | 25.32 ± 0.78 | 18.90 ± 0.57 | 24.45 ± 0.75 | 19.03 ± 0.55 |
| T106 (10/2014) | 23.07 ± 0.73 | 18.59 ± 0.61 | 24.05 ± 0.73 | 19.90 ± 0.64 |
| T124 (11/2016) | 21.86 ± 0.71 | 17.60 ± 0.57 | 22.41 ± 0.71 | 18.12 ± 0.56 |

Values are provided with their associated 1σ errors during pre- and post-cal. These system temperatures are related to the receiving system downstream from its connection with the antenna dish pointed at zenith, and do not account for sky noise entering the antenna (which is the same in the two orthogonally polarized channels). This source of noise is accounted for during the minicals, where the sky noise changes with the ever-changing antenna elevation.

## Amplitude calibration

The computation of the true ratio of orthogonally polarized reflected powers of echoes from Titan's surface to Earth played a pivotal role in our retrieval of correct values of both dielectric constant and small-scale surface roughness. We now discuss briefly the accuracy and possible sources of error in the estimation of the actual level of received power.

The values of the surface dielectric constant were derived from the true ratio of orthogonally polarized reflected powers. The power of Titan's echoes can be computed from the difference between the total signal power received by the antenna and the noise power. When received on Earth at one of the DSN antennas, the complex samples at the output of the radio science receiver (RSR) are attenuated and/or amplified at different steps of the reception chain to increase the SNR and avoid saturation of the analog-to-digital converter (ADC). The exact amplifying gain, in general different for the two orthogonally polarized receiving channels, is unknown a priori, and needs to be estimated to scale the amplified powers down to their true values, which are on the order of $10^{-21}$ Watt (zW).

A typical RSS experiment performed at the closest approach to Titan lasts several hours and is made of three distinct parts: the BSR ingress observation, the radio occultation and the BSR egress observation. Before the ingress the amplifying gain is set, and it remains constant through the entire radio science observation.

The noise power recorded during the observation at the output of the RSR can be expressed as

$$N_{RSR} = k \cdot T_{sys} \cdot B \cdot G_{amp} \tag{8}$$

where k is the Boltzmann constant, $T_{sys}$ is the system noise temperature, B is the receiving bandwidth, and $G_{amp}$ is the total amplifying gain through the reception chain. The variability of system noise temperature drives the noise power variations with time as long as the amplifying gain is stable. Since the system noise temperature mainly depends on the weather, the elevation angle of the receiving DSN antenna, and blackbody radiation from Saturn, it is possible to predict its behavior with time. Instabilities in the amplifying gain would appear as unexpected variations of noise power, not consistent with the previously mentioned phenomena. Such unexpected behavior is not present in the X-band datasets for the T101, T102, T106, and T124 flybys. Only the last of these observations features strong bumps of noise power, but they were recognized to be consistent with rainfalls detected at the DSN complex in Canberra during the observation of

Punga Mare. Hence, we assumed the amplifying gain to be constant throughout each radio science observation.

Real-time measured profiles of system noise temperature are not available for the entire time of observation, but a calibration routine is scheduled before and after each flyby where a bistatic experiment is executed in order to provide enough information to model the noise history (we call these routines respectively Pre-cal and Post-cal). Before Bistatic Ingress and after Bistatic Egress, when the receiving DSN antenna is pointed at the zenith, the RSR connection is switched between the antenna itself, a well-known ambient load and a noise-generating diode nominally working at 12.5 K. A proper analysis of the noise jumps induced by the switches allows for the determination of the actual noise diode temperatures of the two orthogonally polarized channels before and after the entire closest-approach observation. At specific times (usually before Ingress, after Egress, and in-between the two) three short 'minicals' (about 5 minutes) are also scheduled to compute on-the-spot amplifying gain and system noise temperature. Noise diode temperatures serve as inputs to the minicals. For all the flybys we were able to determine diode temperatures $T_{ND}$ for the two channels with less than 1.6 K of uncertainty (Table 2). High values of the RCP diode temperature, with respect to the nominal condition, are consistently observed in all four flybys, both during Pre-cal and Post-cal (NASA operators confirmed this issue in a report from flyby T119). This phenomenon should not represent a problem for our calibration algorithm, and although the RCP diode was working at a higher temperature compared to nominal, there is no reason to expect it to compromise the results of the BSR observations.

Table 3 shows values of inherent receiver temperatures for the four flybys of interest in this paper. These numbers are intermediate results of our calibration routine since the main output of pre-cal and post-cal analysis are the noise diodes' temperatures, which are useful for the minicals analysis. We do not discuss all the details of the calibration process here, as we are planning to include them in a technical document on BSR calibration procedures that is currently in preparation. The main result shown in Table 3 is that the RCP channel is 4–6 K noisier than the LCP channel for all four flybys. The consistency between pre-cal and post-cal outputs, observed also in the noise diodes' temperatures, was expected and strengthens our confidence in the results.

## Phase calibration

Before and after a flyby observation, the Cassini orbiter was scheduled to perform 15 min of free-space baseline, transmitting an unmodulated

carrier directly to Earth. As also observed during the Mars Express mission[36], the received X-RCP carrier always comes with an apparent X-LCP leakage signal about 24 dB weaker. Such a small effect does not impact the computed values of the relative dielectric constant and thus was left uncorrected.

## Equalization

The digitized transfer function of the receiving system is characterized by a spectrum that is not flat at the output of the analog-to-digital converters and produces ripples in the noise floor of the received signal. This may be a relevant issue when processing weak echo components whose amplitude could be on the order of the detected ripples (<2% of the noise pedestal). For completeness, the equalization was applied before computing the noise pedestal and the reflected power coming from Titan's seas. A noise power spectrum was produced by integrating over tens of minutes before or after the BSR observation to accumulate an equalizing spectrum that was later smoothed by a moving average technique. The square root of this was used to scale the voltage spectra of the signal at the output of the RSR. Different equalizing spectra were derived for the two orthogonally polarized channels, but the same small ripple amplitude was detected.

## Model of reflected power

For a proper retrieval of the received power, careful modeling must be carried out independently for the two polarization channels. After choosing the perfectly conducting sphere model and after applying an absolute calibration to the samples, potential power losses through the transmission chain, reception chain, and the medium should be addressed. We are aware that some minor physical phenomena may be missed, but we incorporated as many sources of loss as possible.

For example, we found that reception gain loss due to varying the elevation angle of the Canberra DSN antenna causes a slight change in the amount of received power[61]. Sky conditions at the DSN station can also change throughout the BSR observation and cause slight power losses that are difficult to model. However, qualitative station operators' notes about the average weather during BSR observations and weather condition tables are publicly available. Ranging from a clear to a cloudy sky, we estimated that the difference in power loss would be only about 0.05 dB, which is negligible[61]. We assumed the average clear sky for all the observations and using the formula included in ref. 61 for atmospheric attenuation due to the weather conditions recorded during data reception, we found that the values retrieved are usually <0.5 dB. We applied this correction to the data for the estimation of surface roughness. Note that the T124 observation of Punga was affected by rain at Canberra during reception of the data (≤35 mm/h, ≤75% relative humidity), while during the subsequent observation of the main body of Kraken Mare, acquired about 45 minutes later, the weather was clear (about 0 mm/h and 45% relative humidity). Despite this, we have successfully modeled the increase in noise temperature due to the attenuation from a wet atmosphere (see the comparison between noise level in echoes from Punga and from the other seas in Fig. 3).

Gain losses due to wind loading on the DSN antenna are negligible. We estimated the pointing error of the Cassini antenna with respect to the center of the specular reflection to be <0.1 dB. For this calculation, we used the Ku-band antenna patterns. Because the X-band antenna −3 dB aperture is larger than the Ku-band aperture, we conclude that the pointing error is negligible. We estimated that the pointing accuracy of the Canberra DSN antenna at X-band (0.032 deg) produced losses <0.01 dB and is, therefore, also negligible.

The total power received on the ground from Titan's specular reflections can be modeled using the basic bistatic radar equation[20,35]:

$$P_R = \frac{P_T G_T}{4\pi |T|^2} \sigma \frac{G_R \lambda^2}{(4\pi |R|)^2} \qquad (9)$$

where $P_T$ is the power transmitted by the spacecraft, $G_T$ and $G_R$ are, respectively the gains of the transmitting antenna aboard Cassini and the receiving antenna on Earth (Canberra DSS in our case). These quantities were calibrated by exploiting the 15-minute-long free-space baselines scheduled before and after each BSR experiment, when the spacecraft was transmitting an unmodulated signal directly to Earth. Referring again to Eq. (9), T and R are the distances of transmitter and receiver from the center of Titan, and σ is the bistatic radar cross-section of the observation target, which in this case is the rough surface of a planet. Fjeldbo derived a simple model for the radar cross-section of a rough surface under the Kirchhoff approximations (KA)[35]. If we consider reflectivity to be the ratio between true received power and reflected power expected from a perfectly conducting sphere[20,45], then the radar cross-section can be expressed as:

$$\sigma = \frac{4\pi |T|^2 R_P{}^2 \cos\theta}{(R_P \cos\theta + 2|T'|)(R_P + 2|T'|\cos\theta)} * \Gamma_r \qquad (10)$$

where $R_P$ is the planet radius, θ is the incidence angle, T' is the distance between the transmitter (Cassini) and the specular point, and $\Gamma_r$ is the reflectivity affected by wavelength-scale surface roughness. The reflectivity, which describes how power is redistributed among two orthogonal circular polarization senses after reflection, depends on incidence angle and relative dielectric constant through the Fresnel reflection coefficients[36]. For a random isotropic surface, roughness has the same effect on reflectivity in both polarizations[39], and is modeled with a scale factor function of s:

$$\Gamma_r = \Gamma^* \exp\left\{-4\left(\frac{2\pi}{\lambda} s\cos\theta\right)^2\right\} \qquad (11)$$

where Γ is the reflectivity from a perfectly smooth surface. s can be estimated by inverting the system of formulas reported above for every single received power measurement $P_R$ acquired within each area of interest and using the relative dielectric constant $\varepsilon_r$ obtained from the CPR. From these estimates, we can identify a mean value for s and relative 1σ errors.

## Data availability

The raw RSS data were available from NASA-Planetary Data System (https://pds-atmospheres.nmsu.edu/data_and_services/atmospheres_data/Cassini/Cassini/RSS%20PDS%20page%202019-01-23/rss/TI_10_bis.html). The datasets generated during and/or analyzed during the current study are available from the corresponding author upon request. Source data used for figures and including all the results of the estimations is provided with this paper in the file "Source Data.xlsx". Source data are provided with this paper.

## Code availability

Code useful in order to replicate the results and plots of this study is available on GitHub (https://github.com/NASA-Planetary-Science/rss_ringoccs), where a radio science data reader that can be used to extract the raw data provided by the RSS team is included in the directory rss_ringoccs/rss_ringoccs/rsr_reader. A copy of it is included also in the archive file "Supplementary Software.rar". Note that each bistatic observation of Titan requires its own ad-hoc processing because its timing and geometry depend on which specific flyby it was executed, nevertheless sample code useful to process and calibrate bistatic data is also included in the "Supplementary Software.rar" archive.

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

## Acknowledgements

V.P., A.G.H., D.E.L., and S.M. acknowledge support from NASA CDAP grant NNH21ZDA001N-CDAP. G.B., M.Z., and P.T. acknowledge financial support from the Italian Space Agency through Agreement 2023-6-HH.0, in the context of ESA's JUICE mission. We appreciate the efforts of the Cassini RSS (Radio Science Subsystem) Team in planning and executing these observations and, in particular, we want to express our gratitude to Prof. Essam Marouf for the feedback offered.

## Author contributions

V.P. led and performed the data analysis, conceived the main conceptual ideas, wrote the manuscript, and prepared figures. G.B. performed the data analysis, contributed to the writing of the manuscript, and contributed to the preparation of figures. S.M. helped in decoding the raw data from the PDS node using Python code made available by the RSS team and contributed to the writing of the manuscript. M.Z. contributed to the extraction of the raw BSR data and to the review of the article for intellectual content. L.E.B. helped understand the geometries of the bistatic observations by using SPICE kernels and contributed to the writing of the manuscript. A.G.H, D.E.L., P.D.N., K.O., P.T., J.M.S., and R.D.L. participated in the scientific interpretation of the results and contributed to drafting the article and revising it for intellectual content.

## Competing interests

The authors declare no competing interests.
