## [Peer Review File · Nature Communications]

nature portfolio

Peer Review File

Editorial Note: Parts of this peer review file have been redacted as indicated to avoid any copy right infringement.REVIEWER COMMENTS

Reviewer #1 (Remarks to the Author):

Review of manuscript "Surface properties of Titan's seas as revealed by Cassini RSS bistatic radar experiments"

This manuscript presents an analysis of the dielectric properties and surface roughness of Titan's seas and coastal regions at 3.1 cm wavelength as observed by the Cassini mission in bistatic configuration. The inferred dielectric properties show variability across different seas and are (for the most part) consistent with previous inferences from monostatic data. The surface roughness at millimeter scales was found to be negligible for the seas and comparatively higher for the coastal areas. The surface roughness estimates vary from previous altimeter-derived roughness suggesting dynamic small-scale topography over Titan's seas.

The manuscript is clearly written and organized. The authors have done a good job of making the science accessible to those outside the Saturn / Titan community. In particular, the parallels drawn between coastal landforms on Earth and Titan make it easy to understand Titan specific processes. I recommend the manuscript for publication following some very minor revisions to help readers make the connection between the bistatic analyses and the interpretations of the results from the analyses.

- Please add a few words explaining why an upper bound of 1Hz for spectral broadening is suggestive of suggest smooth surfaces over 10s of meters.
- Figure 1. The figure caption isn't completely informative of what is being shown in the figure. Please add a description of what is represented in the x-axis.
- The purpose of the paragraph "The almost monochromatic ..." is not immediately clear. A topical sentence for this paragraph explaining its significance would be helpful.
- I'm not sure how the Punga Mare permittivity supports the lack of ethane. Some of the other estimated dielectric permittivity values (for L, K1B and K2B) are smaller than the estimates for Punga Mare. So why are only Punga Mare's dielectric properties discussed as being indicative of a lack of ethane?
- Are the differences in uncertainties in the dielectric permittivity reported primarily due to differences in incidence angles between the observations?
- Figure 2. The permittivity and roughness scalebars and the values are not clearly visible. Please make them bigger.

- I don't think mathematical notations like σ_h is explained anywhere except in the table? But maybe I just missed it.
- "However, note that the minimum permissible values of the relative dielectric constant able to produce the observed received powers according to the model in the case of a flat sea surface are always less than 0.06 larger than what we measured." – Can that authors rephrase this sentence. It is hard to understand what the second half of the statement is conveying.

Methods

- In a lot of the rocky planet literature, we see specular reflection from surfaces resulting in the power in the opposite sense circular channel dominating over the signal returns in the same sense circular polarization channel. I would have expected LCP returns to be significantly higher than RCP for coherent, specular reflection from the smooth seas. So, the results presented in Figure 2 are somewhat surprising. Is this an effect of the incidence angle being close to or larger than the Brewster angle? Could the authors add a line to the manuscript to address this.
- There are other multiple scattering mechanisms that could also lead to $CPR = 1$. Why is it that the authors only discuss the impact of the incidence angle on achieving CPR close to unity?
- It is not just the bistatic data, but rather the bistatic dual-polarization data that is making it possible to distinguish roughness from dielectric properties. This is a very cool aspect and could be emphasized further.
- Maybe use a different mathematical notation for radar cross section and the surface roughness since both of them are denoted using σ ?
- Please provide references for the statement that "Surface roughness has the same effect on reflectivity in both polarizations"

Data availability statement

- As we move towards FAIR data sharing, I am disappointed to still see data availability statements that request people to contact the first authors for data needed to reproduce the results presented. It would be so much more beneficial to the community if the authors were to make their study more accessible to everyone by providing access to all the data and software needed for end to end analysis of bistatic, dual-polarization data illustrated here. I would assume that CDAP requires funded work to share data publicly but I could be wrong.

Supplementary information

- The information presented in Figure S1 is not clear. Please revise the x-axis labels and figure captions to explain the information being presented.

Reviewer #2 (Remarks to the Author):

General remarks to authors:

While I am not nearly as familiar with Cassini's RSS instrument as I am with CIRS or VIMS, I do know from just reading the abstract that with single polarization radar measurements, one cannot separate the effects of surface roughness and composition, but with the incorporation of RSS bistatic radar observations, you are then able to make direct measurements of these two parameters. Even without being an expert in this field, I already know that this is a major advancement in adding to the current state of the field and from that alone, this manuscript is worthy of publication. Given that this manuscript presents the first detailed investigation of the bistatic dataset for Titan, thus representing a huge leap forward in enhancing the science return of the Cassini mission, I recommend this manuscript be published after minor revisions.

Note that it would make the review process more efficient if the authors ensured that the manuscript contained both page numbers and line numbers; otherwise, it is more difficult to flush out specific areas during the review process.

Introduction:

PDF page 1, first two sentences in first paragraph: there should be many citations here.

PDF page 1, first paragraph: I doubt refs [1], [2], and [3] are the only published work efforts regarding these sentences. You should change the refs to [e.g., 1], for example, or add more citations.

PDF page 2, Figure caption 1: make sure to redefine the acronyms RCP, LCP, and DSN in the figure caption to make it easier on the readers that might not be familiar with those acronyms (think students).

Dataset:

PDF page 2, first paragraph: if there were 13 bistatic data sets acquired between 2006 and 2016 then why only present 4 in this paper? A sentence on this is needed since you brought it up early in the Introduction. My best guess is there were only 4 datasets acquired of the N. polar seas. But clarification in the manuscript is needed.

PDF page 2, Figure 2: This is an excellent figure, but it is difficult to absorb – it is too busy with all the small text and the amount of text. Consider turning this into a 4-panel figure, 1 panel for each flyby. That way, it'll be much easier to see Cassini's path per flyby. For each flyby curve, perhaps thicken the curves so they stand out easier. Looking at it now, I'm not entirely sure of what the main point of this figure is because there is a lot going on in it. For the color-coded labels of the dielectric constant/roughness, the color bars are hard to see – consider moving them to the side of each panel so they are the same length along the y-axis as for Panels a and b.

Results:

Why doesn't the "Methods" Section precede the "Results" Section? Is this a NCOMMS format requirement? Moving Methods out of order introduces a disconnect to the flow of the paper.

PDF page 3, last sentence in first paragraph: sentence should read "... surfaces far from coastal areas (>~20 km)" or you could say "... surfaces >~20 km from coastal areas."

PDF page 5, paragraph 6: I have some concerns about the statement regarding the weak correlation between dielectric constant and incidence angle. I plotted these two parameters based on your Table 1 values (see attached png). To describe this figure, there is a clear anti-correlation between the two parameters. Do you have any observations of the same footprint but at 2 different angles? That way, you could definitively claim that the correlation is not an issue. Otherwise, you should show the covariance matrix and/or discuss in more detail the degree of anti-correlation between the two parameters. This seems important given that later in the manuscript (PDF page 13) you state that the reflectivity depends on both incidence angle and relative dielectric constant. Since these two parameters are anti-correlated, how can you assure the derived reflectivity values are trustworthy? You should also quantify the statement that no systematic biases are present.

Discussion

PDF page 7, first paragraph: This paragraph is more about motivation/Introduction material for this manuscript. This is important because it justifies the need to analyze the RSS bistatic observations. Consider merging some of it into the Introduction Section since this is the big science reason/need to

do this work. If you leave it in the Discussion Section, then the impact of its importance is lost. If anything, you could open this Section with paragraph 2 and then move paragraph 1 to Introduction.

Methods

PDF page 10, towards the bottom: What is meant by “The history of the system noise temperature....?” Do you mean the variability of system noise over time?

PDF Page 11, Table M1: curious as to why only 1-sigma associated errors are being given. Same comment applies to Table M2.

Dear Reviewers,

The comments you provided really helped us in improving our paper and for this reason we want to thank you very much. We revised our paper in accord with your comments and suggestions. We should have addressed all the points raised and all necessary elements for the evaluation of our work should be present. We will remain available in case more details will be needed, or questions should arise.

Hereafter we will provide a list of the changes made to the paper (they also appear highlighted in red in the text) and then we will answer your comments on a point-by-point base.

List of improvements made to the paper to answer the reviewers or improve its clarity:

Abstract

14. Added 'effective';

15-16. Reworded in 'Results obtained on estuaries suggest...'

Introduction

25-34. Added a bunch of references as requested.

35-51. Paragraph previously contained in the Discussion section moved here.

54. 3.6cm changed in 3.56cm to add more precision

68-70 reworded to answer the reviewer

Dataset

86-87. Added a sentence to answer the reviewer

104-105. rewording of part of the caption (note that all the sigma_h in the paper have been changed to s to answer the reviewer).

109-112. Added reworded to answer the reviewer.

118-119. reworded to answer the reviewer

130-132. added to answer the reviewer

145-147. added to answer the reviewer

154-156. We agree with the reviewer and we deleted the sentence "The values derived herein are similar and support the absence of ethane within Punga Mare" and reworded the previous sentence.

163-174. Reworded and added text to answer the reviewer.

177-180. Slightly reworded to improve understanding.

187-189. Reworded to answer the reviewer.

Table 1. A few decimal digits have been slightly refined

(GS1 1.55->1.56, GS2 1.56->1.57, P 3.5->3.3 mm, K1B 1.3->1.2 mm,

K5 1.7->1.6 mm, K6 3.3->3.2, K8 1.72->1.71, K8 3.2->3.1 mm)

Discussion

225-233. Added text to answer the reviewer concerns about the model we use.

Method

347. Reworded to answer the reviewer.

450. Added text to answer the reviewer

Data Availability

458-463. Text added to answer the reviewer

Supplementary material

Caption Figure S1. Added text to answer the reviewer

Answers to to Reviewer 1:

- Please add a few words explaining why an upper bound of 1Hz for spectral broadening is suggestive of suggest smooth surfaces over 10s of meters.

Thanks for this question, we added text saying that large-scale surface roughness produces a broadening of the received echoes and that in our case the spectral width is so small that (comparable to the resolution) that only a flat surface at the spatial scale of the Fresnel footprint (10s of meters) could produce such a limited frequency spread).

- Figure 1. The figure caption isn't completely informative of what is being shown in the figure. Please add a description of what is represented in the x-axis.

Here the caption of the figure has been rewritten to improve its clarity and highlight the scope of the figure, plus we made clear the meaning of the acronyms we used.

- The purpose of the paragraph "The almost monochromatic ..." is not immediately clear. A topical sentence for this paragraph explaining its significance would be helpful.

Great point, here we added a sentence that clarify the importance of this paragraph. The antenna was able to illuminate the whole area scattering signal back to the radar, thus no underestimation of the received power could have been done.

- I'm not sure how the Punga Mare permittivity supports the lack of ethane. Some of the other estimated dielectric permittivity values (for L, K1B and K2B) are smaller than the estimates for Punga Mare. So why are only Punga Mare's dielectric properties discussed as being indicative of a lack of ethane?

Great point again, for Punga Mare the altimeter (probing liquid column) and the bistatic radar (probing the surface) agree on the value of estimated dielectric constant but drawing general conclusions on the composition of this sea in comparison with the others is challenging. The result obtained by the bistatic observation alone does not support the lack of ethane in Punga, thus, accordingly, we deleted the sentence "The values derived herein are similar and support the absence of ethane within Punga Mare".

- Are the differences in uncertainties in the dielectric permittivity reported primarily due to differences in incidence angles between the observations?

No, they are primarily due to variations in noise level and/or heterogeneities in the sea surface. We added a sentence to make this clearer.

- Figure 2. The permittivity and roughness scalebars and the values are not clearly visible. Please make them bigger.

We agree. We improved Figure 2 by increasing the dimension of labels and colorbars.

- I don't think mathematical notations like σ_h is explained anywhere except in the table? But maybe I just missed it.

Right, first we substituted σ_h with s for the reader not to get confused with σ (radar cross section) and then we included a definition for it in the main text at line 178.

- " However, note that the minimum permissible values of the relative dielectric constant able to produce the observed received powers according to the model in the case of a flat sea surface are always less than 0.06 larger than what we measured." – Can that authors rephrase this sentence. It is hard to understand what the second half of the statement is conveying.

Yes. We reworded the sentence trying to make it clearer: "However, note that a slight increase in the relative dielectric constant (< 0.06 , and thus comparable with estimation errors) would be enough to match the received power if the sea surface was flat at the time of the observation."

Methods

- In a lot of the rocky planet literature, we see specular reflection from surfaces resulting in the power in the opposite sense circular channel dominating over the signal returns in the same sense circular polarization channel. I would have expected LCP returns to be significantly higher than RCP for coherent, specular reflection from the smooth seas. So, the results presented in Figure 2 are somewhat surprising. Is this an effect of the incidence angle being close to or larger than the Brewster angle? Could the authors add a line to the manuscript to address this.

-Certainly. This is exactly an effect of incidence angle being close to or larger than the Brewster angle: hereafter, we report Figures 7 and 8 by [Simpson, 1981], where it is visible that at high incidence angles the same sense component is much more powerful of its opposite. In those bistatic radar experiments made with the Viking on Mars, the observation conditions were very similar to ours.

[redacted]

- There are other multiple scattering mechanisms that could also lead to CPR = 1. Why is it that the authors only discuss the impact of the incidence angle on achieving CPR close to unity?

The main reason is that Titan's seas are expected, from previous studies, to be a well-mixed ternary mixture of liquid hydrocarbons with little or no suspended particles, and the incidence angles we are probing are within a limited range around the Brewster angle at which no surface penetration is expected. Moreover, as we say in the paper, the expressions for the Fresnel reflection coefficients account for the angle of observation when relating relative dielectric constant and reflectivity. Certainly, far from the Brewster's angle the power computation in the weaker polarization may introduce uncertainty in the ϵ_r estimates but still, no systematic bias should be present. As we report in the paper, it is also possible that a thin millimetric surface layer could form on top of the seas and slightly influence our dielectric constant results: this a very interesting hypothesis because from previous studies this layer is not expected to exist. Coherently, we include a new paragraph (lines 225-233) reporting what was found by [Simpson, 1981]. An apparent underestimation of ϵ_r in the Viking observation of Planum Boreum on Mars on 25 Feb 1978 with the specular point going toward the north pole was explained by the presence of CO₂ or H₂O snows covering the northern polar region with surfaces characterized by gradually lower densities than bare soils.

- It is not just the bistatic data, but rather the bistatic dual-polarization data that is making it possible to distinguish roughness from dielectric properties. This is a very cool aspect and could be emphasized further.

Right. We reworded lines 67-70 in the introduction section: "Herein, we present a brief overview of the Cassini RSS bistatic dataset used for this study and provide independent estimates of effective relative dielectric constant and small-scale roughness of Titan's sea surfaces, achieved by exploiting the dual-polarized nature of BSR experiments."

- Maybe use a different mathematical notation for radar cross section and the surface roughness since both of them are denoted using sigma?

Done.

- Please provide references for the statement that "Surface roughness has the same effect on reflectivity in both polarizations"

For the sentence reported at lines 449-450 I added a few words at the beginning pointing out that we work in the hypothesis of a random isotropic surface and I added a reference to the book by Ulaby (2014) where this is clearly explained.

Data availability statement

- As we move towards FAIR data sharing, I am disappointed to still see data availability statements that request people to contact the first authors for data needed to reproduce the results presented. It would be so much more beneficial to the community if the authors were to make their study more accessible to everyone by providing access to all the data and software needed for end to end analysis of bistatic, dual-polarization data illustrated here. I would assume that CDAP requires funded work to share data publicly but I could be wrong.

Good point. We have no problem to share the exact code used to produce each these results. Just note that this is a very complex dataset to analyze because every flyby requires ad-hoc adjustments at all the various steps of processing that should be clear enough from our paper. I added a link to the Data availability section of the paper to the GitHub webpage where an RSR data reader of the bistatic raw data is publicly available, and I also provided an example of Matlab code, with all the steps of processing included, that could be used for processing the data.

Supplementary information

- The information presented in Figure S1 is not clear. Please revise the x-axis labels and figure captions to explain the information being presented.

Yes, we rewrote the caption explaining that this is just a simplified version of the polar stereographic projection reported in Figure 2 of the paper. In order to simplify even more the figure, we also got rid of the scale bars associated to the projection (redundant information already reported in Figure 2).

Answers to Reviewer 2:

- Note that it would make the review process more efficient if the authors ensured that the manuscript contained both page numbers and line numbers; otherwise, it is more difficult to flush out specific areas during the review process.

Thanks for the advice, we added line numbers and we'll be sure to do it also for future submissions.

Introduction:

- PDF page 1, first two sentences in first paragraph: there should be many citations here.
We added a bunch of new citations in the first paragraph as requested.
- PDF page 1, first paragraph: I doubt refs [1], [2], and [3] are the only published work efforts regarding these sentences. You should change the refs to [e.g., 1], for example, or add more citations.

Added more citations and 'e.g.' as suggested.

- PDF page 2, Figure caption 1: make sure to redefine the acronyms RCP, LCP, and DSN in the figure caption to make it easier on the readers that might not be familiar with those acronyms (think students).

Here the caption of the figure has been rewritten to improve its clarity and highlight the scope of the figure, plus we made clear the meaning of the acronyms we used.

Dataset:

- PDF page 2, first paragraph: if there were 13 bistatic data sets acquired between 2006 and 2016 then why only present 4 in this paper? A sentence on this is needed since you brought it up early in the Introduction. My best guess is there were only 4 datasets acquired of the N. polar seas. But clarification in the manuscript is needed.

Exactly. We added a sentence at lines 86-87 to explain that we limited our study to these four flybys because these are the only observations whose specular point crossed the main body of at least one of the three polar seas (Kraken, Ligeia, and Punga Mare).

- PDF page 2, Figure 2: This is an excellent figure, but it is difficult to absorb – it is too busy with all the small text and the amount of text. Consider turning this into a 4-panel figure, 1 panel for each flyby. That way, it'll be much easier to see Cassini's path per flyby. For each flyby curve, perhaps thicken the curves so they stand out easier. Looking at it now, I'm not entirely sure of what the main point of this figure is because there is a lot going on in it. For the color-coded labels of the dielectric constant/roughness, the color bars are hard to see – consider moving them to the side of each panel so they are the same length along the y-axis as for Panels a and b.

This is a Figure that works both as a context map for showing the location of the bistatic tracks as well as a presentation of the results obtained. We agreed that the figure could be improved for better clarity, so we increased the dimension of labels and color bars as suggested.

Results:

- Why doesn't the "Methods" Section precede the "Results" Section? Is this a NCOMMS format requirement? Moving Methods out of order introduces a disconnect to the flow of the paper.

Yes, this is the format of Nature Communications.

- PDF page 3, last sentence in first paragraph: sentence should read "... surfaces far from coastal areas (>~20 km)" or you could say "... surfaces >~20 km from coastal areas."

Agreed, we changed it in "...far from coastal areas (>~20 km)" as suggested.

- PDF page 5, paragraph 6: I have some concerns about the statement regarding the weak correlation between dielectric constant and incidence angle. I plotted these two parameters based on your Table 1 values (see attached png). To describe this figure, there is a clear anti-correlation between the two parameters. Do you have any observations of the same footprint but at 2 different angles? That way, you could definitively claim that the correlation is not an issue. Otherwise, you should show the covariance matrix and/or discuss in more detail the degree of anti-correlation between the two parameters. This seems important given that later in the manuscript (PDF page 13) you state that the reflectivity depends on both incidence angle and relative dielectric constant. Since these two parameters are anti-correlated, how can you assure the derived reflectivity values are trustworthy? You should also quantify the statement that no systematic biases are present.

This is a very good point. Unluckily we do not have observations of the same footprint but at 2 different angles so we decided: first, to add a sentence to point out what is the degree of anti-correlation between the two parameters (lines 164-165); second, to add the paragraph at lines 168-174 in which we show that previous experimental investigations conducted on Earth at X-band showed how the model that we adopted perform very well when the surface small-scale roughness is so small as the one that we have on Titan's seas; third, to add the paragraph at lines 225-233 where we report about a measurement made by [Simpson, 1981]. An apparent underestimation and incidence angle correlation of ϵ_r , recorded during the Viking observation of Planum Boreum happened on Mars on 25 Feb 1978, with the specular point moving toward the north pole, that was successfully explained by the presence of CO₂ or H₂O snows covering the northern polar region with surfaces characterized by gradually lower densities than bare soils. Simulations were also performed by [Simpson, 1981], that confirmed the good performances of the model we used for our paper.

Discussion

- PDF page 7, first paragraph: This paragraph is more about motivation/Introduction material for this manuscript. This is important because it justifies the need to analyze the RSS bistatic observations. Consider merging some of it into the Introduction Section since this is the big science reason/need to do this work. If you leave it in the Discussion Section, then the impact of its importance is lost. If anything, you could open this Section with paragraph 2 and then move paragraph 1 to Introduction.

Thank you for this suggestion that really improves the clarity of the paper. We moved the paragraph in the Introduction section.

Methods

- PDF page 10, towards the bottom: What is meant by "The history of the system noise temperature....?" Do you mean the variability of system noise over time?

Yes, we reworded the sentence starting at line 346 in this way: "The variability of system noise temperature..."

- PDF Page 11, Table M1: curious as to why only 1-sigma associated errors are being given. Same comment applies to Table M2.

We made this choice to follow what we found in literature. In particular, papers previously published by Simpson et al. usually report results with their associated standard deviations.

REVIEWERS' COMMENTS

Reviewer #1 (Remarks to the Author):

All my previous comments have been adequately addressed in the revised version of the manuscript. I have two minor comments regarding the flow of the revised paper for the authors to consider. Otherwise, the initial paper was very well-organized and so is this revised version. I would also like to thank the authors for taking steps to make their data and analysis accessible to the community. This is a neat study that has wrangled some complex Cassini observations into useful interpretations that complement existing radar studies of Titan. I look forward to the paper being published and wish the authors the best.

Page 1, Line 35. The second paragraph of the introduction (which was moved here from the discussion) reads more like an abrupt insertion of text rather than a well-organized merger with the existing introduction. For instance, at the end of paragraph one, the authors mention a “previously unexploited dataset”, but it is now followed up with a discussion on the Cassini SAR instrument instead of the Cassini RSR. Please consider moving a few sentences around or rewording some of the text to re-establish the flow that was present prior to this revision.

Page 8, Line 225. While the addition of the paragraph about Viking data is helpful in addressing some of the concerns raised by the reviewers, from the perspective of someone reading the published article without insights into the review process, the Viking paragraph might read like an unnecessary jump to Mars in the midst of discussing Titan. I will ultimately leave this up to the authors but one potential way to address this would be to move the Viking text to Line 253 where the implausible presence of a thin surface layer on Titan is discussed. The Viking bistatic experiments could be used simply as an example of previous interpretations of low dielectric permittivity to represent a porous / lower density surface layer, similar to how the lunar regolith is currently discussed.

Reviewer #2 (Remarks to the Author):

Thank you for taking the time to address all of my reviewer comments. I am happy with the changes made and I recommend the manuscript for publication.

Dear Reviewers,

Once again thanks for all your comments. We revised our paper to address the remarks of Reviewer #1 and the editorial requests that we found on the author checklist. We should have addressed all the points raised. While addressing the editorial requests, we realized that the paper cannot contain a section entitled "Dataset". To comply with Nature Communications requests, we had to move that section right before the ending paragraph of the Introduction section. We think that this doesn't really alter too much the flow of the paper. Hereafter we will provide a list of the changes made to the paper (they also appear highlighted in red in the text).

REVIEWERS' COMMENTS

Reviewer #1 (Remarks to the Author):

All my previous comments have been adequately addressed in the revised version of the manuscript. I have two minor comments regarding the flow of the revised paper for the authors to consider. Otherwise, the initial paper was very well-organized and so is this revised version. I would also like to thank the authors for taking steps to make their data and analysis accessible to the community. This is a neat study that has wrangled some complex Cassini observations into useful interpretations that complement existing radar studies of Titan. I look forward to the paper being published and wish the authors the best.

Thank you very much!

Page 1, Line 35. The second paragraph of the introduction (which was moved here from the discussion) reads more like an abrupt insertion of text rather than a well-organized merger with the existing introduction. For instance, at the end of paragraph one, the authors mention a "previously unexploited dataset", but it is now followed up with a discussion on the Cassini SAR instrument instead of the Cassini RSR. Please consider moving a few sentences around or rewording some of the text to re-establish the flow that was present prior to this revision.

We agree: we reworded some of the text before and after the second paragraph to re-establish the lost flow.

Page 8, Line 225. While the addition of the paragraph about Viking data is helpful in addressing some of the concerns raised by the reviewers, from the perspective of someone reading the published article without insights into the review process, the Viking paragraph might read like an unnecessary jump to Mars in the midst of discussing Titan. I will ultimately leave this up to the authors but one potential way to address this would be to move the Viking text to Line 253 where the implausible presence of a thin surface layer on Titan is discussed. The Viking bistatic experiments could be used simply as an example of previous interpretations of low dielectric permittivity to represent a porous / lower density surface layer, similar to how the lunar regolith is currently discussed.

We agree: we moved the Viking text at line 253 close to where we mention the result of the lunar regolith.

Reviewer #2 (Remarks to the Author):

Thank you for taking the time to address all of my reviewer comments. I am happy with the changes made and I recommend the manuscript for publication.

Great, thank you!

List of changes made to the paper to answer the reviewers and the editorial requests:

9. Added Corresponding author

Introduction

35. Added sentence.

53. Added sentence.

74-91. Dataset section was moved here.

99-102. A new sentence has been added to answer an editorial request (*"Please begin the final paragraph of Introduction from "Herein, we present a brief overview" and modify that part to introduce briefly the analysis and the main findings."*)

Table 1

Corrected a couple of typos in the number of samples for the Areas K2 and K3.

Discussion

231. Paragraph about Viking was moved down, at line 253-261. Added few words to adjust to the change at lines 231-233 and 262. List of the references and reference numbers have been updated coherently with the change.